# On the Inherent Instability of Biocognition: Toward New Probability Models and Statistical Tools

**DOI:** 10.3390/e24081070

**Published:** 2022-08-03

**Authors:** Rodrick Wallace, Irina Leonova, Saikat Gochhait

**Affiliations:** 1The New York State Psychiatric Institute, 1051 Riverside Dr, New York, NY 10032, USA; rodrick.wallace@nyspi.columbia.edu; 2Faculty of Social Sciences, Lobachevsky University, 603950 Nizhny Novgorod, Russia; irina.leonova@unn.ru; 3Neuroscience Research Institute, Samara State Medical University, 89 Chapaevskaya str., 443001 Samara, Russia; 4Symbiosis Institute of Digital and Telecom Management, Symbiosis International Deemed University, Symbiosis Knowledge Village, Village- Lavale, Tahasil- Mulshi, Pune 412115, India

**Keywords:** cognition, control theory, distortion, information theory, phase change, rate distortion function, stochastic stability

## Abstract

A central conundrum enshrouds biocognition: almost all such phenomena are inherently unstable and must be constantly controlled by external regulatory machinery to ensure proper function, in much the same sense that blood pressure and the ‘stream of consciousness’ require persistent delicate regulation for the survival of higher organisms. Here, we derive the Data Rate Theorem of control theory that characterizes such instability via the Rate Distortion Theorem of information theory for adiabatically stationary nonergodic systems. We then outline a novel approach to building new statistical tools for data analysis based on those theorems, focusing on groupoid symmetry-breaking phase transitions characterized by Fisher Zero analogs.

## 1. Introduction

Although, as Maturana and Varela [1] put it, all organisms—with and without a nervous system—are cognitive, and living per se is a process of cognition, the assertion, as it were, only sings half the Mass. Many, perhaps most, cognitive phenomena, from blood pressure control through gene expression, immune function, consciousness, and driving a fast vehicle at night on a twisting, pot-holed roadway, are inherently unstable and must be heavily regulated to prevent catastrophe. For humans in particular, the ‘stream of consciousness’ must always be contained within social and cultural as well as more common neuropsychological ‘riverbanks’ to successfully confront and circumvent powerful selection pressures (e.g., [2]).

Cognition implies choice, choice reduces uncertainty, and the reduction in uncertainty implies the existence of an information source ‘dual’ to the cognitive process under study [2,3,4,5,6]. The argument is both direct and unambiguous. Here, we will assume such information sources are piecewise adiabatically stationary, so that on a trajectory ‘piece’, the system is sufficiently close to nonergodic stationary for the appropriate asymptotic limit theorems to work sufficiently well. A useful analog is the Born–Oppenheimer approximation in molecular QM, where nuclear motions are taken as slow enough for electron structures to essentially equilibrate.

Relatively recent work relates the asymptotic limit theorems of information theory to control theory via the Data Rate Theorem (e.g., [7]). Following the linear approximation used in that work, consider a reduced model of a control system as in Figure 1.

An initial *n*-dimensional vector of system parameters at time *t* is represented as $x_{t}$. The system state at time t+1 is then—near a presumed nonequilibrium steady state—approximated by the first-order relation:(1)$$x_{t+1}\approx Ax_{t}+Bu_{t}+W_{t}$$
where A and B are fixed *n*-dimensional square matrices, $u_{t}$ is a vector of control information, and $W_{t}$ is an *n*-dimensional vector of Brownian white noise. According to the Data Rate Theorem, if H is a rate of control information sufficient to stabilize an inherently unstable control system, then it must be greater than a minimum, $H_{0}$,
(2)$$H>H_{0}\equiv \log(∥\det\left[A^{m}\right]∥)$$
where det is the determinant of the subcomponent $A^{m}$ – with m≤n – of the matrix A having eigenvalues ≥1. $H_{0}$ is defined as the rate at which the unstable system generates its own ‘topological information’.

If this inequality is violated, stability fails: blood pressure skyrockets, the immune system eats its host, conscious higher animals suffer hallucinations, and a speeding vehicle goes off the road.

A singular feature of such a control theory perspective, as a reviewer pointed out, is that not only does it separate a car and driver from the road, but it differentiates a road from cars and drivers, which may be of considerable importance if one’s central focus is highway maintenance. Further as a reviewer has noted, the channel can be rendered noise-free by importing more information through expanding the environment—the controller—so that the system/controller interaction occurs at the boundary that separates them. This is, however, seldom possible in real-world, real-time interactions.

Here, we will approach these matters from a somewhat unusual direction that permits significant generalization and engineering application, in a large sense. We will derive the Data Rate Theorem from the Rate Distortion Theorem via some counterintuitive but characteristic formal legerdemain.

## 2. Remembering Brownian Motion

Einstein’s treatment of Brownian motion for *N* particles, via the simple diffusion equation
(3)$$\partial \rho \left(x,t\right)/\partial t=\mu \partial^{2}\rho \left(x,t\right)/\partial x^{2}$$
where *x* is distance and μ a diffusion coefficient, produces a solution in terms of the Normal Distribution,
(4)$$\rho \left(x,t\right)=\frac{N}{\sqrt{4\pi \mu t}}\exp\left[-x^{2}/\left(4\mu t\right)\right]$$

After some manipulation,
(5)$$\sqrt{<x^{2}>}\propto \sqrt{t}$$

A central observation of Bennett, as represented by Feynman [8], is that a ‘simple ideal machine’ permits the extraction of free energy from an information source. The Rate Distortion Theorem [9] expresses the minimum channel capacity R(D) needed for a transmitted signal along a noisy channel to be received with average distortion less than or equal to some scalar measure *D*. A ‘worst case’, in many respects, is the Gaussian channel for which, under the square distortion measure,
(6)$$\begin{array}{c} R\left(D\right)=\frac{1}{2}\log_{2}\left(\sigma^{2}/D\right),\;\;\;D\leq \sigma^{2}\\ R\left(D\right)=0,\;\;\;D>\sigma^{2} \end{array}$$
where σ is a ‘noise’ parameter.

If, following Feynman [8], we identify information as a form of free energy, it becomes possible—*in a purely formal manner*—to construct an ‘entropy’ *S* for the system in the standard way of ordinary thermodynamics via the Legendre transform
(7)S≡−R(D)+DdR/dD
and to then impose a first-order nonequilibrium thermodynamics Onsager approximation [10] as the simple diffusion equation
(8)$$dD/dt\approx \mu dS/dD=\mu Dd^{2}R/dD^{2}$$
where ‘μ’ is taken as a kind of diffusion coefficient. Using Equation (Equation 6), we can directly calculate D(t) as
(9)$$\begin{array}{c} dD/dt=\frac{\mu }{2D(t)\log(2)}\\ D\left(t\right)\propto \sqrt{t} \end{array}$$
Thus, based on the definition of ‘entropy’ in Equation (Equation 7) as a Legendre transform and the use of the Onsager entropy gradient approximation of Equation (Equation 8), $D\left(t\right)\propto \sqrt{t}$, *as if the ‘target’ undergoes simple Brownian motion* in the absence of further data. Parenthetically, as will be shown below in Section 4, this surprising result suggests a direction for generalization to less simple systems, i.e., build an appropriate free energy, define an ‘entropy’ as its Legendre transform, and then impose the Onsager entropy gradient approximation to derive system dynamics. Taking the next step in the full argument, for a simple deterministic system, we can impose ‘control free energy’ at a rate M(H), depending, in a monotonic increasing but possibly nonlinear manner, on a control information rate H. Then, for the existence of a nonequilibrium steady state—a kind of stability having, at least, a fixed value of distortion— dD/dt must decline to zero,
(10)$$\begin{array}{c} dD/dt=\mu Dd^{2}R/dD^{2}-M\left(H\right)\leq 0\\ M\left(H\right)\geq \mu Dd^{2}R/dD^{2}\geq 0\\ H\geq H_{0}\equiv \max\left\{M^{-1}\left(\mu Dd^{2}R/dD^{2}\right)\right\}\} \end{array}$$
where max represents the maximum value, and noting that, if *M* is monotonic increasing, so is the inverse function $M^{-1}$. By convexity, $d^{2}R/dD^{2}\geq 0$ [9]. For the Gaussian channel—in the absence of further noise—at nonequilibrium steady state where dD/dt≡0, then D∝1/M(H).

Again, an example might be continuous radar or lidar illumination of a moving target with transmitted energy rate *M*, defining a maximum *D* inversely proportional to *M*. If illumination fails, then *D* will increase as $\sqrt{t}$, representing classic diffusion from the original tracking trajectory.

We have, in the last expression of Equation (Equation 10), replicated something of Equation (Equation 2), but there is far more structure concealed here.

## 3. A General Model

Recall again that the Rate Distortion Function R(D) is always convex in *D*, so that $d^{2}R/dD^{2}\geq 0$ [9,11]. Further, following [11,12], and others, for a nonergodic process, the Rate Distortion Function can be calculated as an average across the RDF of the ergodic components of that process, and can thus be expected to remain convex in *D*. We reconsider Figure 1 and now examine the ‘expected transmission’ of a signal $X_{t}\to {\hat{X}}_{t+1}$, (in the presence of an added ‘noise’ Ω), but received as a de facto signal $X_{t+1}$. That is, there will be some deviation between what is ordered and what is observed, measured under a scalar distortion metric as $d({\hat{X}}_{t+1},X_{t+1})$ and averaged as
(11)$$D=\sum Pr\left({\hat{X}}_{t+1}\right)d\left({\hat{X}}_{t+1},X_{t+1}\right)$$
where Pr is the probability of ${\hat{X}}_{t+1}$ and the ‘sum’ may represent a generalized integral. We have constructed an adiabatically, piecewise stationary information channel for the control system and can invoke a Rate Distortion Function R(D). Following Equation (Equation 10), we can write a general stochastic differential equation [13]
(12)$$dD_{t}=\left(\mu D_{t}{\left[d^{2}R/dD^{2}\right]}_{t}-M\left(H\right)\right)dt+\Omega h\left(D_{t}\right)dW_{t}$$
where μ is a ‘diffusion coefficient’ and Ωh(D) is the ‘volatility’ under Brownian noise $dW_{t}$. As above, an average <D> can be calculated from the relation
$$<dD_{t}>=\mu D_{t}{\left[d^{2}R/dD^{2}\right]}_{t}-M\left(H\right)=0$$
so that, again, something like Equation (Equation 10) can be said to hold. However, stability of stochastic systems is a far richer landscape than for deterministic systems. That is, we are now interested in stability under stochastic volatility, and there are many possible characterizations of it. More specifically, we want to calculate the nonequilibrium steady-state properties of a general function $Q(D_{t})$ given Equation (Equation 11), i.e., the relation $<dQ_{t}>=0$. This can be conducted using the Ito Chain Rule [13], leading to
(13)$$\left(\mu Dd^{2}R/dD^{2}-M\left(H\right)\right)dQ/dD+\frac{\Omega^{2}}{2}h{\left(D\right)}^{2}d^{2}Q/dD^{2}=0$$

Solving for M(H),
(14)$$M\left(H\right)=\mu D\left(\frac{d^{2}}{dD^{2}}R\;\left(D\right)\right)+\frac{\Omega^{2}h\;\left(D\right)\left(\frac{d^{2}}{dD^{2}}Q\;\left(D\right)\right)}{2\left(\frac{d}{dD}Q\;\left(D\right)\right)}$$
giving general results that are not confined to a particular algebraic form for R(D).

As a ‘worst case’ example, we treat the Gaussian channel of Equation (Equation 6) in second order.

Assuming the simple volatility function h(D)=D and taking $Q\left(D\right)=D^{2}$ allows determination of a general condition for stability in variance for the Gaussian channel from Equation (Equation 14) as
(15)$$\begin{array}{c} M\left(H\right)=\frac{\Omega^{2}}{2}D+\frac{1}{D\log(4)}\geq \frac{\Omega }{\sqrt{\log(2)}}\\ H\geq M^{-1}\left(\frac{\Omega }{\sqrt{\log(2)}}\right) \end{array}$$
where the last inequality in the first expression can be found from a simple minimization argument. Calculations for other kinds of channels are similar.

Another possible approach views the control signal H itself as the fundamental distortion measure between intended and observed behaviors, so that Equation (Equation 11) becomes
(16)$$H\equiv \sum Pr\left({\hat{X}}_{t+1}\right)H\left({\hat{X}}_{t+1},X_{t+1}\right)$$
where H is the rate of control information needed to stabilize the inherently unstable system and the other factors are as above. We can now carry through the analysis as driven by Equation (Equation 12), but with *D* replaced by *H* and the rate of control free energy is taken as M(H). This gives the necessary condition of Equation (Equation 14), but in terms of *H*, it provides a different picture of stability dynamics based on the ‘average control distortion’ *H* rather than on H itself, regardless of the typically monotonic increasing nature of M(H).

## 4. Extending the Perspective

The focus on distortion and channel capacity in our reinterpretation of Figure 1 permits a simplified one-dimensional analysis. We continue in one dimension but branch the argument.

Via the Legendre transform and Onsager approximations of Equations (Equation 7) and (Equation 8), and through the SDE of Equation (Equation 12), it has been possible to derive, in Equations (Equation 14) and (Equation 16), a fair version of the Data Rate Theorem result of Equation (Equation 2). A constraint on the approach is that, because information transmission is not microreversible—for example, in English, the term ‘the’ has a much higher probability than ‘eht’, and in Chess, one cannot uncheckmate a King—there can be no ‘Onsager reciprocal relations’ in multidimensional variants of Equations (Equation 8) and (Equation 12).

The treatment of multiple parallel processes—branching—in this formalism is of particular interest if somewhat subtle.

Here, we see control dynamics—the set of possible ‘Data Rate Theorems’—as highly context-dependent, supposing that different ‘road conditions’ will require different analogs to, or forms of, the DRT. That is, the transmission of control messages as represented in Equation (Equation 11) involves equivalence classes of possible control sequences $X\equiv \{X_{t},\;X_{t+1},\;X_{t+2},\;\ldots \}$. For example, driving a particular stretch of road slowly on a dry, sunny morning is a different ‘game’ than driving it at high speed during a midnight snowstorm.

We characterize all possible such ‘games’ in terms of equivalence classes $G_{j}$ of the control path sequences *X*, each class associated with a particular ‘game’ represented by an information source having uncertainty $H_{G_{j}}\equiv H_{j}$.

In this model, R(D), characterizing the particular ‘vehicle’, plays a different role. Here, we fix the maximum acceptable average distortion and impose the corresponding *R* as a temperature analog via an iterated model. To do this, we construct a pseudoprobability
(17)$$P_{j}=\frac{\exp[-H_{i}/g\left(R\right)]}{\sum_{j}\exp\left[-H_{i}/g\left(R\right)\right]}$$
where the ‘temperature’ g(R) must be calculated from first principles.

The denominator in Equation (Equation 17) can be taken as a statistical mechanical partition function to derive a free energy *F* as
(18)$$\begin{array}{c} \exp\left[-F/g\left(R\right)\right]=\sum_{k}\exp\left[-H_{k}/g\left(R\right)\right]\equiv A\left(g\left(R\right)\right)\\ F(R)=-\log[A(g(R))]g(R) \end{array}$$

Next, again, it is possible to formally define an ‘entropy’ in terms of a Legendre transform on *F* and to again impose a dynamic relation like that of the first-order Onsager treatment of nonequilibrium thermodynamics [10]. Then, in general (dropping the ‘diffusion coefficient’ μ),
(19)$$\begin{array}{c} S(R)\equiv -F(R)+RdF/dR\\ \partial R/\partial t\approx dS/dR=Rd^{2}F/dR^{2}=f\left(R\right)\\ F\left(R\right)=\int \int \frac{f(R)}{R}dRdR+C_{1}R+C_{2}\\ g\left(R\right)=\frac{-F(R)}{RootOf\left(\exp[Z]-A(-F(R)/Z)\right)} \end{array}$$

Several points:(1)In particular, note the first-order Onsager nonequilibrium thermodynamics approximation in the second expression. This is a very crude model that cannot be expected to have universal applicability. Higher-order and multivariate versions will be of greater, but still limited, use.(2)f(R) represents the ‘friction’ inherent to any control system, e.g., dR/dt=f(R)=β−αR,R(t)→β/α, while the RootOf construction generalizes the Lambert W-function, seen by carrying through a calculation setting A(g(R))=g(R).(3)As ‘RootOf’ may have complex number solutions, the temperature analog g(R) now imposes ‘Fisher Zeros’ analogous to those characterizing the phase transition in physical systems (e.g., [14,15,16]). Phase transitions in the cognitive process range from the punctuated onset of conscious signal detection to Yerkes–Dodson ‘arousal’ dynamics. See Wallace [2] for worked-out examples.(4)The set of equivalence classes $G_{j}$ defines a groupoid in the classic manner (e.g., [17]), and the Fisher Zero construction represents groupoid symmetry-breaking for cognitive phenomena that is analogous to, but different from, the group symmetry-breaking associated with the phase transition in physical processes.

Groupoid symmetry-breaking phase transitions represent one extension of the basic DRT.

(5)One possible extension of these results is via a ‘reaction rate’ treatment abducted from chemical physics [18]. The ‘reaction rate’ *L*, taking the minimum possible source uncertainty across the ‘game’ played by the control system as $H_{0}$, is then
(20)$$L=\frac{\sum_{H_{j}\geq H_{0}}\exp\left[-H_{j}/g\left(R\right)\right]}{\sum_{k}\exp\left[-H_{k}/g\left(R\right)\right]}\equiv L\left(H_{0},g\left(R\right)\right)$$
where the sums may be generalized integrals and the denominator can be recognized as the partition function A(g(R)) from Equation (Equation 18).

Wallace [2] uses similar models to derive a version of the Yerkes–Dodson ‘inverted-U’ model for cognition rate vs. arousal.

Again, Fisher Zero analogs in g(R) characterize cognition rate phase transitions.

(6)We have, by careful design, been able to restrict the development to one dimension via focus on the Rate Distortion Function R(D) alone. The extension of the theory to higher dimensions, for example, writing R(D,Z), where Z is an irreducible vector of resource rates, is not entirely straightforward [19] (Chapter 5), nor is the extension to more complicated versions of the Onsager models [20].(7)The argument extends directly to stationary nonergodic systems, where time averages are not ensemble averages, assuming that sequences can be broken into small high-probability sets consonant with underlying forms of grammar and syntax, and a much larger set of low-probability sequences not so consonant [2,19,20]. Equation (Equation 17) is then path-by-path, as source uncertainties can still be defined for individual paths [21]. The ‘game’ equivalence classes still emerge directly, leading to groupoid symmetry-breaking phase transitions [2,22].

Finally, it is possible to explore the stochastic stability of the system of Equation (Equation 19) in much the same manner as was performed in the previous section. Here, the driving stochastic relation—analogous to Equation (Equation 12)—becomes
(21)$$dR_{r}=f\left(R_{t}\right)dt+\Omega h\left(R_{t}\right)dW_{t}$$
where, again, Ωh(R) is a volatility term in the Brownian noise $dW_{t}$.

The application of the Ito Chain Rule to a stochastic stability function Q(R) produces an analog to Equation (Equation 13), $<dQ_{t}>=0$, as
(22)$$f\left(R_{t}\right)\frac{d}{dR_{t}}Q\left(R_{t}\right)+\frac{\Omega^{2}{\left(h\left(R_{t}\right)\right)}^{2}\frac{d^{2}}{d{R_{t}}^{2}}Q\left(R_{t}\right)}{2}=0$$

Examining stability in second order and with ‘simple’ volatility, so that $Q=R^{2}$ and h(R)=R, and assigning an ‘exponential’ relation as dR/dt=f(R)=β−αR gives the variance as
(23)$$Var={\left(\frac{\beta }{\alpha -\Omega /2}\right)}^{2}-{\left(\beta /\alpha \right)}^{2}$$
which explodes as $\Omega^{2}\to 2\alpha$.

A different viewpoint permits *R* in Equation (Equation 22) to be unconstrained, thus imposing selection pressure via the possible forms of f(R) and *F*, as determined by *Q* and *h*.

## 5. Discussion

The ‘simple’ Rate Distortion Theorem arguments of Section 2 and Section 3, driven by the essential convexity of the Rate Distortion Function for stationary systems, lead to very general deterministic and stochastic forms of the Data Rate Theorem. That theorem, usually derived as an extension of the Bode Integral Theorem (e.g., [7], and references therein), places control theory and information theory within the same milieu, one that encompasses much of the cognitive phenomena so uniquely characterizing the living state (e.g., [1]).

More generally, there cannot be cognition—held within the confines of information theory—without a parallel regulation, held within the confines of control theory. Both theories are constrained by powerful—and apparently closely related—asymptotic limit theorems.

Section 4 extends the underlying concept to multiple ‘selection pressures’ via equivalence classes of control path sequences. This leads to an iterated model in which a partition function, the denominator of Equation (Equation 17), serves as the basis for an iterated free energy, its associated Legendre transform entropy, and an iterated Onsager approximation in the gradient of that entropy construct, driving system dynamics. A stochastic stability analysis can be conducted in a standard manner. Ultimately, the extended theory focuses on groupoid symmetry-breaking phase transitions characterized by Fisher Zero analogs.

In sum, we have outlined a novel approach to building new statistical tools for data analysis based on the asymptotic limit theorems of control and information theories. An interested reader should be able to take this material and run with it, recognizing that all such statistical tools—much like Onsager approximations in nonequilibrium thermodynamics—inevitably have limited ranges of reliability and applicability. Such recognition serves to evade many of the intellectual thickets surrounding contemporary Grand Unifying Theories, e.g., [23,24,25,26,27].

## Figures and Tables

**Figure 1 entropy-24-01070-f001:**
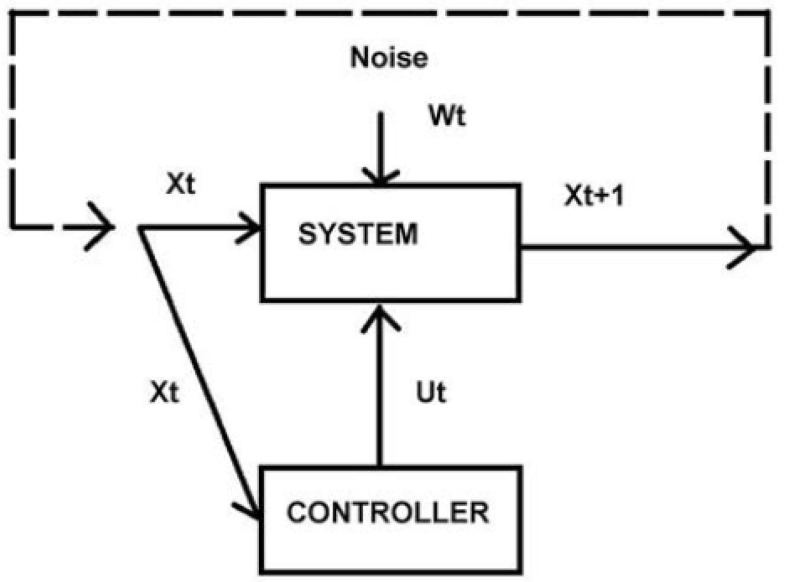
A simplified model of an inherently unstable system stabilized by a control signal $U_{t}$. A singular feature of this perspective, as a reviewer has pointed out, is that not only does it separate the car and driver from the road, but it differentiates the road from the car and driver, which may be of considerable importance if one’s focus is highway maintenance.

## Data Availability

Not Applicable.

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
