# Peer review of "On the Inherent Instability of Biocognition: Toward New Probability Models and Statistical Tools"

_entropy, 2022, doi:10.3390/e24081070_

Round 1

Reviewer 1 Report

General remark: The authors derive the Data Rate theorem of control theory that characterize instability observed in biocognition via the rate distortion theory of information theory. Then they outline a novel approach to building new statistical data analysis tools. The manuscript has mild issues and required authors to resolve them before I can recommend it for publication in Entropy.

Issues

It requires more elaboration on the “entropy”, Eq. (7). For instance, what does it mean in this framework? A measure of channel capacity? It will further strengthen the logic of imposing Onsager approximation in Eq. (8).

Furthermore, it also requires more elaboration on Eq. (9). For instance, whether control free energy M has an inverse. Besides, what’s the reasoning behind this assumption?

1   The manuscript lacks proper definitions for some variables although it may not play  important roles in derivation. It will increase readability.

a.       The x and mu in Eq. (3)

b.      The sigma in Eq. (6).

2    There is a typo in Eq. (20).

Author Response

a). The argument on ‘entropy’ – Eqs.(7) to (10) – has been strengthened as recommended.
b). The control free energy M in Eq.(10) will be monotonic in the control signal \mathcal{H}, but not necessarily linear: think of a driver wrestling with a manual steering wheel on a pot-holed road. Seeing the next pot hole ahead instantiates the control signal, which can be subsequently back-estimated by the amount of force the driver uses to avoid it. Yes, M has an inverse.
c). Descriptions have been added to Eqs.(3) and (8) as recommended. Eq.(20) has been corrected, and the argument in Section 3 clarified

Reviewer 2 Report

This is a compact and well-written paper making a simple but very important and often-neglected point: any system that interacts with an environment uses its inputs from (the non-noise components of) its environment to maintain a form of quasi-stability, in particular, to maintain the functional distinction between itself and its environment.  This applies equally to the "system" and the "controller" in Fig. 1, which in fact interact symmetrically.  To use the authors' example of dangerous driving, the car-driver system must use each input from the road to predict (i.e. adjust its behavior to suit) the road's next state sufficiently accurately to avoid becoming no longer distinguishable, as a functioning dynamical system, from the road.  The road faces a similar problem: it must adjust what it does to the car-driver system with sufficient accuracy to avoid becoming no longer distinguishable, as a functional dynamical system, from what remains of the car and driver.

Karl Friston and colleagues call this sort of response to control inputs "active inference," but sometimes fail to point out the symmetry of the situation.  See K. Friston, arxiv:1906.10184 .

It is worth noting that the channel can always be rendered noise-free by expanding the environment (i.e. the controller) so that the system-controller interaction occurs at the boundary that separates them.

Sect. 4 is very nice, as it addresses the question of what happens when the system-environment interaction has and can switch between different "sectors" with qualitatively different behavior.

Author Response

a). We have taken the liberty of incorporated some of the comments into the text, with credit to the reviewer.
b). The last paragraph of the Discussion places this work in a somewhat different context from Grand Unifying Theories.
